# Effect of Temperature, Water Activity and Incubation Time on Trichothecene Production by *Fusarium cerealis* Isolated from Durum Wheat Grains

**DOI:** 10.3390/pathogens12050736

**Published:** 2023-05-19

**Authors:** Jessica G. Erazo, Sofía A. Palacios, Nuria A. Veliz, Agostina Del Canto, Silvana Plem, María L. Ramirez, Adriana M. Torres

**Affiliations:** Instituto de Investigación en Micología y Micotoxicología (IMICO), Consejo Nacional de Investigaciones Científicas y Tecnológicas (CONICET), Universidad Nacional de Río Cuarto (UNRC), Río Cuarto 5800, Argentina; jerazo@exa.unrc.edu.ar (J.G.E.); spalacios@exa.unrc.edu.ar (S.A.P.); nuriaaveliz@gmail.com (N.A.V.);

**Keywords:** nivalenol, deoxynivalenol, ecophysiology, growth, *Fusarium*, durum wheat

## Abstract

*Fusarium cerealis* is a causal agent of *Fusarium* Head Blight in wheat, and it produces both deoxynivalenol (DON) and nivalenol (NIV). Nevertheless, the effect of environmental factors on the growth and mycotoxin production of this species has not been studied so far. The objective of this study was to investigate the impact of environmental factors on the growth and mycotoxin production of *F. cerealis* strains. All strains were able to grow in a wide range of water activity (a_W_) and temperatures, but their mycotoxin production was influenced by strain and environmental factors. NIV was produced at high a_W_ and temperatures, while optimal conditions for DON production were observed at low a_W_. Interestingly, some strains were able to simultaneously produce both toxins, which could pose a more significant risk for grain contamination.

## 1. Introduction

*Fusarium* head blight (FHB) is an important fungal disease affecting wheat production worldwide. Economic losses occur due to reductions in grain yield and quality, mainly from the degradation of gluten proteins by the fungus resulting in lower bakery properties [1]. In addition, grains could be contaminated with mycotoxins that, in some cases, are regulated at the maximum permitted levels, making cereal trading more difficult. In addition, these mycotoxins are harmful for human and animal health [2]. The main mycotoxins produced are trichothecenes which are sesquiterpenoides that have in common a tricyclic 12,13-epoxytrichothec-9-ene (EPT) core structure [3]. According to the substitution groups on the EPT core, they are divided into four classes (types A–D). Deoxynivalenol (DON) and nivalenol (NIV) are included within the type B trichothecenes. These mycotoxins affect protein synthesis and, in consequence, DNA biosynthesis by inhibiting the peptidyl transferase activity of 60S ribosomes [4]. In humans and animals, exposure to mycotoxins causes acute and chronic effects such as abdominal pain, feed refusal, anorexia, diarrhea, immunological problems, vomiting, skin dermatitis, and hemorrhagic lesions [5,6]. Trichothecenes can act as immunostimulatory or immunosuppressive, depending on the dose, exposure frequency, and type of immune function assay [7]. In plants, they can cause dwarfism, chlorosis, and inhibition of root and shoot elongation and act as a virulence factor in FHB [8]. DON is the main trichothecene found in cereals and cereal-based products [9]. However, it has been reported that NIV is more cytotoxic than DON [10,11,12]. In addition, several reports showed additive and/or synergetic effects when DON is combined with NIV; however, these interactions are dose-dependent [13,14,15]. For instance, Alassane-Kpembi et al. [16] showed that below a cytotoxicity level of 50%, the combination of DON and NIV causes a synergistic effect. This suggests that low doses of both toxins may have greater toxicity than each mycotoxin alone, when they are found simultaneously in food and feed.

FHB is caused, mainly, by species within the *Fusarium graminearum* species complex, with *F. graminearum sensu stricto* being the most-reported causal agent. Some other *Fusarium* species such as *F. culmorum*, *F. poae*, *F. avenaceum*, and *F. cerealis* can also cause the disease [2]. Recently, *F. cerealis* (synonym *F. crookwellense*) was reported causing the disease in wheat, barley, and oat in different parts of the world [17,18,19,20,21,22,23,24]. In the field, the *Fusarium* inoculum remains on crop debris as ascospores within perithecia (sexual structures) or macroconidia, and there it can resist the environmental conditions. Under favorable weather conditions (high relative humidity and warm temperatures) during wheat anthesis, the inoculum is capable of dispersing by wind or rain to reach the anther and start the plant infection. First, the spores germinate; then the hyphae grow on the ovary, palea, and lemma; and, after that, they start mycotoxin production [25]. Gil-Serna et al. [23] reported that a shift in *Fusarium* species might be occurring in oats cultivated in Spain, since *F. cerealis* was isolated from contaminated samples while the main causal agent, *F. graminearum* was not detected. In wheat, *F. cerealis* causes a negative impact on germination percentage and a reduction in the length and weight of coleptiles; it can cause crown rot and produce severe FHB symptoms and grains contaminated with mycotoxins, mainly type B trichothecenes [26,27]. Moreover, it is frequently isolated from maize, where it can cause ear rot and stalk rot [28,29], and it has been associated with root rot of soybean and ginseng [30,31].

In a previous study, we isolated *F. cerealis* from symptomatic durum wheat grains obtained from the main producing area of Argentina, the south of the Buenos Aires Province, during four harvest seasons [32]. In addition, in a few of these samples, we detected NIV contamination, some of which had high levels (2380–8190 µg/kg, unpublished data) as well as DON contamination [33,34]. In a more recent published paper, we reported that *F. cerealis* strains were capable of producing both DON and NIV toxins in planta and in vitro. Moreover, the strains were able to produce 22 other metabolites, including zearalelone (ZEA), its derivatives α and β zearalenol, and the so-called “emerging mycotoxins” [35]: enniatins (ENNs), beauvericin (BEA), fusaric acid, culmorin, and butenolide [27]. Taking this into account, durum wheat grains contaminated with this species pose a risk to human and animal health. However, the environmental conditions leading to the production of DON or NIV or their simultaneous production have not been determined yet. Fungal growth and trichothecene biosynthesis by the *Fusarium* species depend on complex interactions between biotic and abiotic factors. Within the latter, temperature and water activity (a_W_) represent the main factors that have a strong influence on the transcriptional activation of mycotoxin biosynthetic genes and mycotoxin production [36].

These factors were extensively studied in other FHB pathogens and trichothecene type B producers such as *F. graminearum* and *F. culmorum* [37,38,39,40]; however, there is a lack of ecophysiological studies about *F. cerealis*. Hence, the objective of this study was to determine the impact of a_W_, temperature, and time on growth and DON and NIV production by three *F. cerealis* strains on durum wheat grain-based media. These strains were selected by considering their different mycotoxin production profiles on durum wheat grain cultures [27], in order to cover all the intraspecific variabilities.

## 2. Materials and Methods

### 2.1. Fungal Strains

*Fusarium cerealis* strains RCFG 6029, RCFG 6046, and RCFG 6076 were isolated from FHB symptomatic durum wheat grains from the main growing area in Argentina. The identity of the RCFG 6029 strain was confirmed by sequencing analysis of EF-1α gene (Accession N° KX359404), while the identity of the remaining strains was confirmed by using species-specific primers and inter-simple sequence repeats (ISSR) analysis [32,41]. Moreover, a phenotypic and molecular characterization of their mycotoxin production profile was previously carried out. The three strains produced DON and NIV on durum wheat grain cultures: the RCFG 6029 strain produced 16 µg/kg of NIV and 160 µg/kg of DON, the RCFG 6046 strain produced 193 µg/kg of NIV and 270 µg/kg of DON, and the RCFG 6076 strain produced 9284 µg/kg of NIV and 170 µg/kg of DON [27]. Cultures were purified by subculturing single macroconidia and preserved in sterile 15% glycerol at −80 °C [42]. Preserved cultures are maintained in the culture collection of the Research Institute on Mycology and Mycotoxicology (IMICO), CONICET-UNRC.

### 2.2. Medium

Durum wheat was finely milled by using a Romer mill (Romer Labs Inc., Union, MO, USA). Mixtures of 2% (*w*/*v*) of milled wheat in water were prepared, and 2% (*w*/*v*) agar (technical agar N° 2, Oxoid, Waltham, MA, USA) was added. The a_W_ of the basic medium was adjusted to 0.99, 0.97, 0.95, 0.93, and 0.90 by addition of different amounts of glycerol [43]. The media were autoclaved at 120 °C for 20 min. Flasks of molten media were thoroughly shaken, prior to being poured into 9 cm sterile Petri dishes. The a_W_ of representative samples (2 of each treatment) of media was checked with an Aqualab Series 3 (Decagon Devices, Inc., Pullman, WA, USA). Additional, uninoculated control plates were prepared and measured at the end of the experiment in order to detect any significant deviation of the a_W_.

### 2.3. Inoculation, Incubation, and Growth Assessment

Petri plates were inoculated with a 4 mm diameter agar disk that was taken from the margin of a 7-day-old colony of each isolate grown on synthetic nutrient agar [42] at 25 °C and transferred face down to the center of each plate. Inoculated plates of the same a_W_ were sealed in polyethylene bags and incubated at 15, 20, 25, and 30 °C for 28 days. At 7, 14, 21, and 28 days after incubation, some plates were taken in triplicate for mycotoxin analysis. A full factorial design was used, where the factors were a_W_, temperature, and strain, and the response was growth (total number of plates: 5 a_W_ × 4 temperatures × 4 incubation periods × 3 strains × 3 replicates).

Assessment of growth was made every day during the incubation period, and two diameters of the growing colonies were measured at right angles to each other until the colony reached the edge of the plate. Colonies’ radii were plotted against time, and linear regression was applied in order to obtain the growth rate (mm/day) as the slope of the line. After the incubation period, uninoculated controls and treatments were frozen for later mycotoxin determination.

### 2.4. Nivalenol and Deoxynivalenol Analysis

For mycotoxin analysis, three replicates per treatment were destructively sampled after 7, 14, 21, and 28 days. DON and NIV extraction was completed according to Li et al. [44], with some modifications. Briefly, 8 g of culture medium were shaken with 20 mL of acetonitrile in a vortexing machine (Vortexer) for 5 min and then on an orbital shaker (150 rpm) for 1 h. Subsequently, the sample was centrifuged at 10,000× *g* rpm for 5 min (Sorvall™ ST 16, Thermo Fisher Scientific, Waltham, MA, USA). Next, 8 mL of the supernatant were transferred into a 15 mL centrifuge tube containing 1.5 of NaCl. The tube was shaken for 2 min by using a vortexing machine. Then, it was centrifuged at 3500 rpm for 5 min, and 5 mL of the supernatant were transferred into a vial. The acetonitrile was blown to dryness under nitrogen flow. The dried residue was redissolved in 1 mL (acetonitrile/water = 1:9, *v*/*v*) and homogenized with a vortexing machine, and then a nylon filter (0.22 μm) was applied to remove impurities. Samples were stored at 4 °C until analysis by HPLC. The HPLC system consisted of a Waters e2695 separations module (Milford, MA, USA) with a stainless steel C18 reversed-phase column (150 × 4.6 mm, 5 μm particle size; LunaPhenomenex) connected to pre-column (20 × 4.6 mm, 5 μm particle size, Luna-Phenomenex). Mycotoxins were detected by UV at 220 nm using a Waters 2998 diode array detector. The mobile phase was water:methanol (88:12, *v*/*v*) at a flow rate of 1.3 mL/min. The limit of detection (LOD) was 15 µg/kg, and the limit of quantification (LOQ) was 40 µg/kg. The retention times of NIV and DON were 5.2 min and 10.7 min, respectively.

### 2.5. Statistical Analysis

The growth rate and mycotoxin concentration data were transformed to normal scores, and they were evaluated by analysis of variance (ANOVA) using InfoStat for Windows version 2018 [45]. For normality and homogeneity of variances, data were tested using Shapiro–Wilk test and Levene test, respectively. Statistical significance was judged at the level of *p ≤* 0.0001. When the analysis was statistically significant, the post hoc Fisher’s least significant difference (LSD) test was used for the separation of means (significance judged at *p ≤* 0.05).

## 3. Results

### 3.1. Effect of a_W_ and Temperature on Lag Phase

The ANOVA analysis (Table 1) shows that all the single variables and two- and three-way interactions had a significant impact on the lag phase of the three strains, except the a_W_ × strain interaction. The Table 1 reveals that the a_W_ influenced the lag phase (F = 860.95) the most, followed by temperature and strain.

Figure 1A–C show that the shortest lag phases were observed at 25 °C, 0.99 a_W._ They corresponded to the *F. cerealis* RCFG 6076 (23.47 h) and *F. cerealis* RCFG 6046 (24.86 h) strains. In general, the lag phases increased at a more stressful a_W_ (0.93–0.95) and low temperatures (15–20 °C). However, the behavior of the RCFG 6076 strain showed some differences, since at 15 °C the lag phase was relatively short and did not show significant differences at a_W_ ≥ 0.97. These values were similar to those obtained at 30 °C, 0.95 and 0.97 a_W_. The largest lag phase (107.74 h) was observed for the RCFG 6029 strain at 15 °C, 0.93 a_W_ (Figure 1A).

### 3.2. Effect of a_W_ and Temperature on Growth Rate

Table 1 shows that *F. cerealis* growth rate was significantly influenced by all the single factors evaluated in the study (strain, a_W_, and temperature) and by their interactions. The a_W_ had the most important effect on the growth rate. Figure 1D–F show the growth rates of all strains according to the different a_W_ and temperatures studied. For the three *F. cerealis* strains, the maximum growth was observed at 25–30 °C, 0.99 a_W_, while the minimal growth was observed at 15 °C, 0.93 a_W_. The growth of all strains decreased as the water availability of the media was reduced. No growth was observed at 0.90 a_W_ during the incubation period, regardless of the temperature, for any tested strain.

The highest growth rate was observed for the RCFG 6076 and RCFG 6046 strains at 25 °C, 0.99 a_W_ (8.27 and 8.14 mm/day, respectively), and the values were statistically significantly different compared to the rest of the obtained growth rates. Within some a_W_, temperature showed little or no effect on the growth rates. At 25–30 °C, there were not any statistically significant differences among the growth rates at 0.93, 0.95, and 0.97 a_W_ for the RCFG 6076 strain (Figure 1E), at 0.95 and 0.97 for the RCFG 6029 strain (Figure 1D), and at 0.95 for the RCFG 6046 strain (Figure 1F).

On the other hand, the lowest growth rate was observed for the RCFG 6046 strain at 15 °C, 0.93 a_W_ (0.74 mm/day).

In order to identify the optimum and marginal conditions that allowed the growth of *F. cerealis* strains, three contour maps were performed. They show the isopleths for different growth rates according to a_W_ and temperature. Figure 2 shows that *F. cerealis* can grow in a wide range of a_W_ (0.99 to 0.93) and temperatures (15–30 °C).

### 3.3. Effect of a_W_, Temperature, and Incubation Time on Nivalenol and Deoxynivalenol Production

All the factors under study (a_W_, temperature, incubation time, and strain) and their interactions influenced NIV and DON production by the three *F. cerealis* strains on a wheat-based medium (Table 2). The analysis of variance showed that the most important factor that influenced the NIV production was strain, while, for DON production, the incubation time and a_W_ were the most important. Figure 3 and Figure 4 show the NIV and DON production by *F. cerealis* strains at 15, 20, 25, and 30 °C; at four different a_W_; and during 7, 14, 21, and 28 days of incubation. In general, *F. cerealis* was able to produce DON and NIV at all the conditions tested; however, the amount and the type of toxin was strain-dependent, since they showed different trichothecene production profiles.

The toxin levels observed for the RCFG 6076 strain were higher than those observed for the other strains under almost all the temperatures and a_W_ that were tested. The highest NIV level (23,918 µg/kg) was produced by the RCFG 6076 strain at 30 °C, 0.99 a_W_, after 14 days of incubation. The maximum NIV production of the other strains was 13,432 µg/kg (30 °C, 0.99 a_W_, 14 days) and 3,527 µg/kg (25 °C, 0.99 a_W_, 7 days) for RCFG 6046 and RCFG 6029, respectively. The lowest level detected (41 µg/kg) was observed at 20 °C, 0.93 a_W_, after 28 days of incubation for the RCFG 6076 strain. Before this period, the toxin was not detected in samples at low a_W_ and temperatures (LOD: 15 µg/kg, LOQ: 40 µg/kg) (Figure 3).

In general, the optimal condition for NIV production was 25–30 °C, 0.99 a_W_; as a_W_ was reduced, NIV production declined. For the RCFG 6076 and RCFG 6046 strains, NIV production was higher at 30 °C, but it decreased at 25 °C and 20 °C; while, for the RCFG 6029 strain, higher production was observed at 25 °C, but it decreased at 30 °C. At 15 °C, NIV was only produced by the RCFG 6076 strain at all a_W_ that were tested after 28 days of incubation.

The three *F. cerealis* strains were capable of producing DON; however, the production profile was different for each strain (Figure 4). DON was produced by all the studied strains only after 28 days of incubation. The maximum production (2954 µg/kg) was observed at 30 °C, 0.93 a_W_ (RCFG 6076 strain), while the minimum value obtained was 660 µg/kg at 15 °C, 0.93 a_W_ (RCFG 6046 strain). DON production was mostly observed at a_W_ ≤ 0.97 at all temperatures, although the maximum levels were mainly observed at 0.93 a_W_. At 20 °C, all strains produced DON but at different a_W_. At 15 °C, only the RCFG 6046 strain produced the toxin at 0.93 a_W_. The RCFG 6029 strain only produced DON at 20 °C, 0.99–0.97 a_W_.

Maximum NIV levels were detected at 14 days of incubation, while, for DON, they were detected after 28 days of incubation. Considering this, we analyzed the optimum conditions for temperature and a_W_, where the *F. cerealis* strains produced the maximum levels of NIV and DON, in order to identify those conditions with the highest toxicological risks for each strain (Figure 5). Higher NIV levels were detected at 0.99 a_W_ for the three strains. The RCFG 6029 strain produced the maximum NIV levels at 25 °C; the RCFG 6076 strain at the range of 25–30 °C, and the RCFG 6046 strain at 30 °C. The maximum DON levels produced by RCFG 6029 were observed at 20 °C, and 0.97–0.99 a_W_. For RCFG 6076 strain, the highest levels were observed at 25–30 °C and 0.93–0.95 a_W_. While, for RCFG 6046 the highest production was detected at 25 °C, 0.95 a_W_. However, appreciable levels were also obtained at 15 °C, 0.93 a_W_ (Figure 5).

There were a few conditions where the strains co-produce both toxins (Figure 6). Only strains RCFG 6076 and RCFG 6046 were able to co-produced NIV and DON after 28 days of incubation. The RCFG 6046 strain only produced both toxins at 0.95 a_W_ and 25 °C, while the RCFG 6076 strain co-produced DON and NIV at 0.95 a_W_ and temperatures ≥ 20 °C and at 0.97 a_W_ and 20 °C. In most cases, NIV production was higher than DON production.

## 4. Discussion

In a previous study we have analyzed the mycotoxin profile of *F. cerealis* and we observed that it was capable of producing a wide range of mycotoxins and secondary metabolites including DON and NIV, both in vitro and in planta [27]. However, the environmental factors that influence its growth and trichothecene production have not been determined so far. The present study evaluated for the first time the influence of temperature, a_W_ and time of incubation on *F. cerealis* growth and DON and NIV production, on a wheat-based medium. The three *F. cerealis* strains evaluated were able to grow at 0.99–0.93 a_W_ and 15–30 °C. Maximum growth rates were observed at 0.99 a_W_ and 25–30 °C and the optimal conditions for growth ranged from 0.99 to 0.97 a_W_ for all tested temperatures. Minimal growth rates were observed at 15 °C, 0.93 a_W_. No growth was observed at 0.90 a_W_ regardless of temperature and time of incubation tested. These growth parameters are in agreement with those reported for others FHB pathogens such as *F. graminearum* and *F. culmorum* which grew best at 0.995–25 °C and were able to grow in the range of 0.995–0.93 a_W_ and 15–30 °C. In addition, no mycelial growth was observed at a_W_ ≤ 0.90 [37,38,39,40].

*Fusarium* head blight is greatly affected by weather conditions. Infections during the flowering period are favored by extended periods (2–3 days) of >90% relative humidity and temperatures in the range of 15–30 °C [46]. These conditions are optimal for *Fusarium* pathogen development, as demonstrated in this study for *F. cerealis*.

Regarding mycotoxin analysis, the production patterns observed for the three *F. cerealis* strains were totally different.

Nivalenol was detected in all the conditions in which *F. cerealis* was able to grow. The highest levels were detected at 25–30 °C, 0.99 a_W_, between 14 and 21 days of incubation; after this period, the levels decreased. These temperatures and a_W_ were the same in which the maximum growth was observed for *F. cerealis*, while the minimum levels detected for NIV were at 15 and 20 °C, 0.93 a_W_, after 28 days of incubation. Therefore, *F. cerealis* is capable of producing NIV at suboptimal conditions for growth, but it takes a longer period of time. These same results were observed for other *Fusarium* species such as *F. graminearum* and *F. culmorum*, which produced higher amounts of trichothecene at the highest a_W_ and temperatures analyzed [38,40]. For instance, in Argentina, Ramirez et al. [39] observed that *F. graminearum* produced the maximum amount of DON at 0.995 a_W_ and 30 °C on irradiated wheat grains. For those NIV-producing species such as *F. poae*, *F. meridionale*, *F. culmorum*, and *F. asiaticum*, it was reported that the optimum conditions for toxin production were between 20 and 30 °C and 0.98 and 0.995 a_W_ [37,40,47,48].

Regarding DON production, the three *F. cerealis* strains were capable of producing the toxin, but only after 28 days of incubation. DON was produced at lower levels than NIV. The maximum levels were also obtained at 25–30 °C; however, contrary to NIV production, at 0.95 and 0.93 a_W_. This means that DON production was favored by the stressful conditions related to a_W_. This is in accordance with Schmidt-Heydt et al. [49], who developed a model to predicted DON production by *F. culmorum*, relating the specific gene expression of the key biosynthetic genes with a_W_ levels. This model suggests that high DON levels would occur under water stress. Moreover, Marin et al. [50] observed an induction in the *Tri5* gene expression (implicated in DON biosynthesis) of *F. graminearum* under water stress. Taking this into account, there could be a risk of grain contamination with DON during storage, since, according to Magan et al. [51], fungal spoilage and contamination of grains with mycotoxins may continue during storage if the moisture, temperature, and aeration are suitable for fungal growth and toxin production. They observed that DON contamination on stored wheat inoculated with *F. graminearum* exceeded the maximum permitted level (<1250 ppb) when it was stored at 30 °C and 0.95 and 0.93 a_W._ In addition, Garcia Celá et al. [52] analyzed mycotoxins (included DON and NIV) in naturally contaminated stored wheat and wheat inoculated with *F. graminearum* and concluded that the optimum production of these compounds may occur at 20–25 °C and 0.95 a_W_. They observed that the DON and NIV contamination levels were significantly affected by a_W_.

In addition, the present study shows that it is necessary to include more than one strain in ecophysiological studies to obtain more reliable results, due to the intraspecific variability that exists in a species, mainly within *Fusarium* [53]. For instance, in our study, we observed that, regarding growth conditions, the three strains shared the same patterns. However, when we analyzed the mycotoxin production, they had different production profiles. The RCFG 6076 strain produced the highest mycotoxin levels in all conditions. This is in accordance with our previous work, where we observed that this strain produced the maximum NIV level on durum wheat grain cultures [27]. Moreover, we observed that the RCFG 6029 strain produced DON in completely different conditions compared to the other strains. We could not have observed these results using only one strain.

The co-production of both toxins was observed in a few conditions, most of them for the RCFG 6076 strain, at 0.95 a_W_ and between 20 and 30 °C, after 28 days of incubation. This is in agreement with our previous report of DON and NIV co-production by *F. cerealis* strains on durum wheat grain cultures and in planta. However, when *F. cerealis* strains grew on autoclaved durum wheat grains at 25 °C for 28 days, many of them produced higher amounts of DON than NIV. However, if we consider the two strains under study that co-produce both toxins, RCFG 6076 produced higher amounts of NIV, while RCFG 6046 produced higher amounts of DON [27]. This is consistent with the present study.

## 5. Conclusions

*F. cerealis* strains were capable of producing NIV in a wide range of temperatures and a_W_. High amounts of this toxin were produced at 0.99 a_W_ and 25–30 °C, which are the same environmental conditions for optimal growth. These conditions are needed for *Fusarium* head blight development. So, in years with conducive conditions for the disease, grains could be contaminated with high NIV levels if they are infected by *F. cerealis*. Although DON production was not detected under these conditions, the risk of grain contamination with this mycotoxin could occur under storage conditions, since the optimal production seems to be at low a_W_. Moreover, the toxin was produced after 28 days of incubation by all strains. Why DON is produced after this period of time is a question that we still have to answer. This is, to our knowledge, the first ecophysiological study carried out about *F. cerealis*, so we do not have other studies to compare our results with. In order to answer this question, we are carrying out more studies, especially gene expression studies.

In addition, two strains were able to simultaneously produce both toxins. Therefore, grains could end up contaminated with both DON and NIV if *F. cerealis* is present, posing a more significant risk for human and animal consumption.

## Figures and Tables

**Figure 1 pathogens-12-00736-f001:**
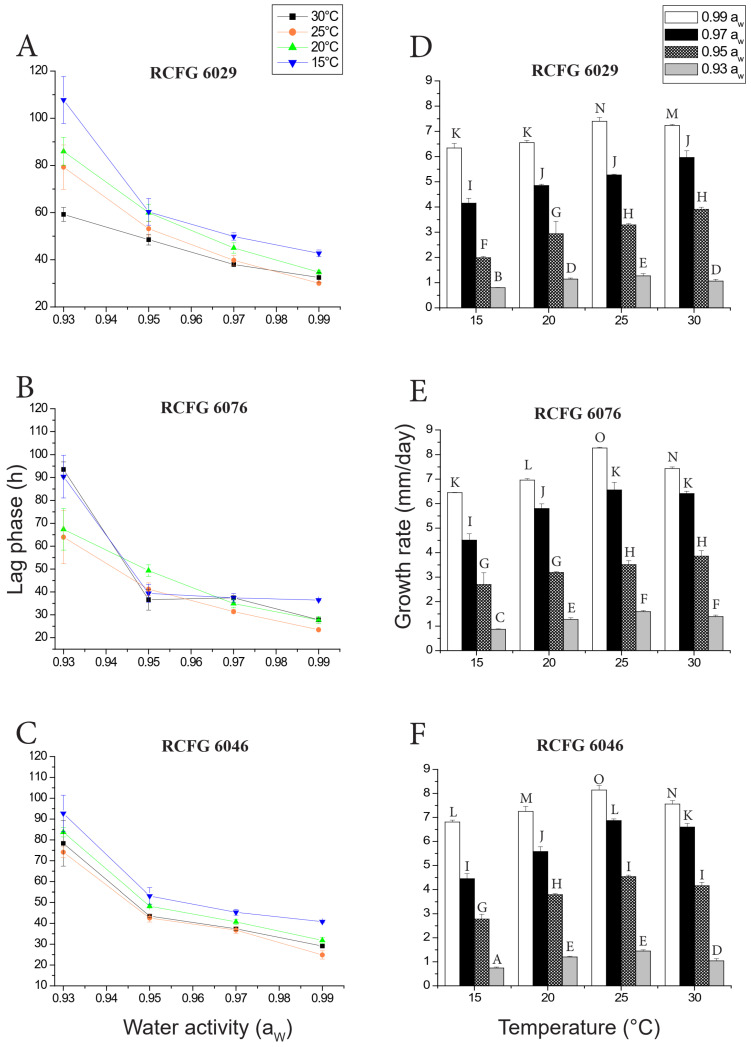
Lag phases (**A**–**C**) and growth rates (**D**–**F**) of three *Fusarium cerealis* strains (RCFG 6029, RCFG 6076, and RCFG 6046) according to water activity (a_W_) (0.93–0.99) and different temperatures (15, 20, 25, and 30 °C) in a wheat-based medium. The error bars indicate the standard deviation for the triplicates. Different letters indicate statistical differences according to LSD test (*p* < 0.05).

**Figure 2 pathogens-12-00736-f002:**
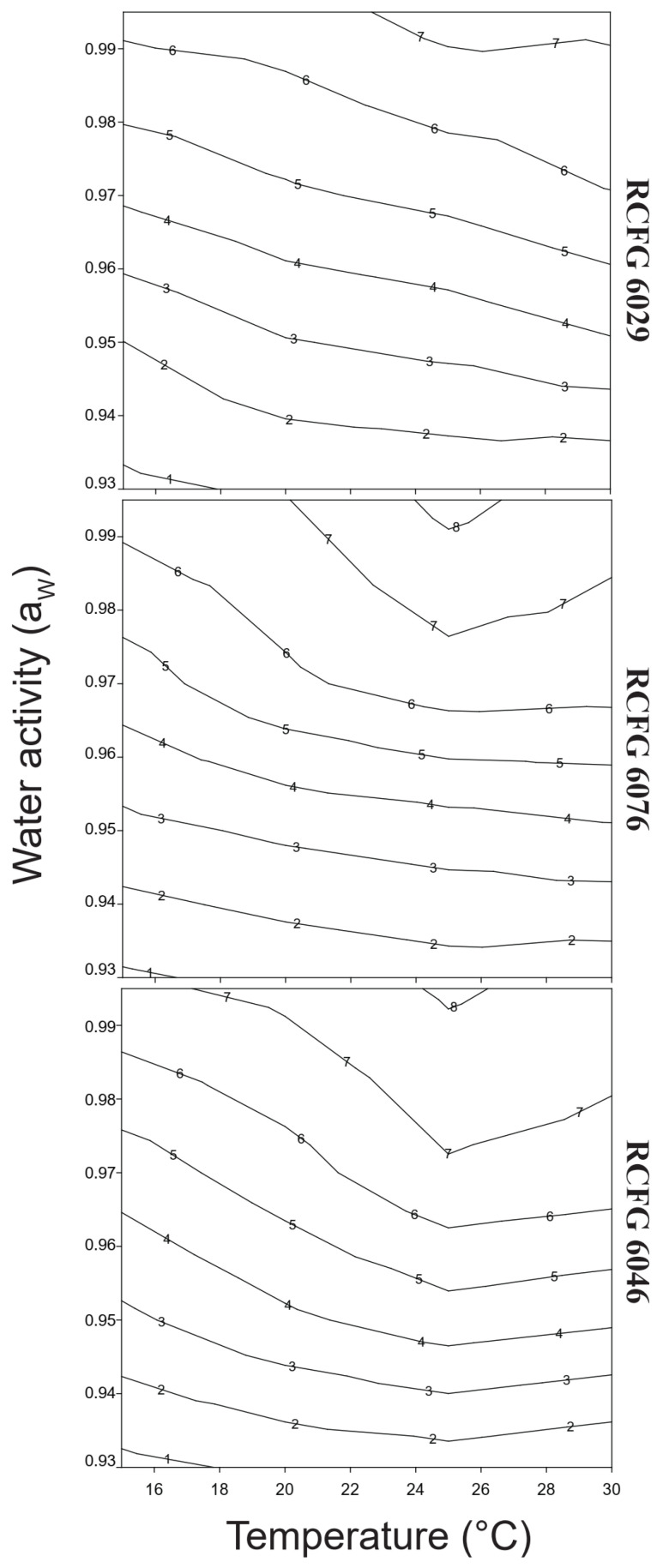
Two-dimensional contour map of growth profile of *Fusarium cerealis* strains (RCFG 6029, RCFG 6076, and RCFG 6046) in relation to water activity (a_W_) (0.93–0.99) and temperatures in a wheat-based medium. The number of isopleths corresponds to similar radial growth rates (mm/day).

**Figure 3 pathogens-12-00736-f003:**
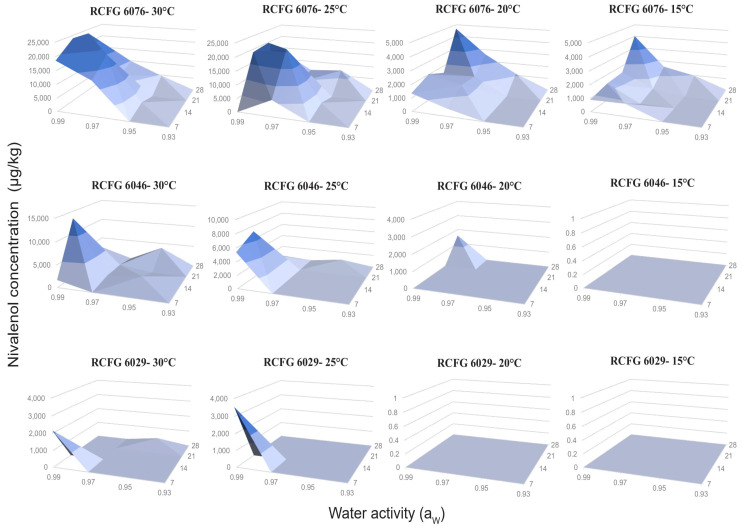
Nivalenol concentrations produced by three *Fusarium cerealis* strains (RCFG 6029, RCFG 6076, and RCFG 6046) according to water activity (a_W_) (0.93–0.99) and different temperatures (15 °C, 20 °C, 25 °C, and 30 °C) in a wheat-based medium during 7, 14, 21, and 28 days of incubation. Limit of detection (LOD): 15 µg/kg, Limit of quantification (LOQ): 40 µg/kg.

**Figure 4 pathogens-12-00736-f004:**
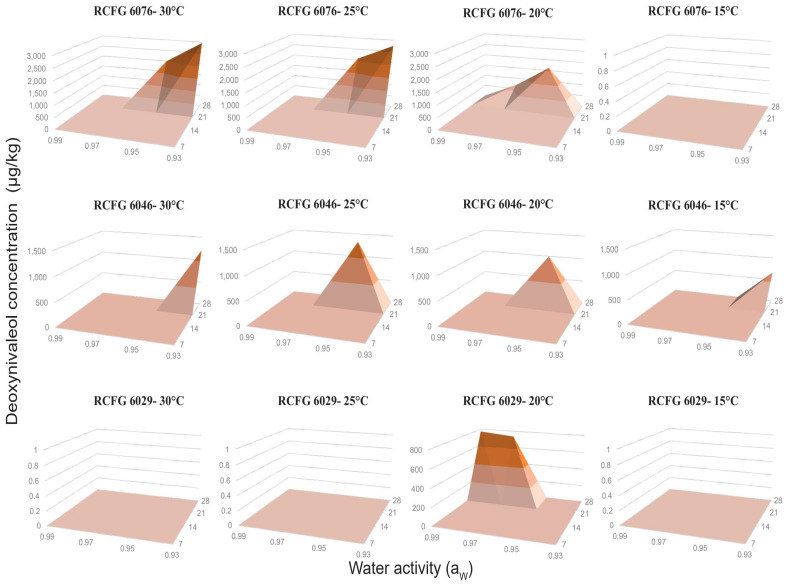
Deoxynivalenol concentrations produced by three *Fusarium cerealis* strains (RCFG 6029, RCFG 6076, and RCFG 6046) according to water activity (a_W_) (0.93–0.99) and different temperatures (15 °C, 20 °C, 25 °C, and 30 °C) in a wheat-based medium during 7, 14, 21, and 28 days of incubation. Limit of detection (LOD): 15 µg/kg, Limit of quantification (LOQ): 40 µg/kg.

**Figure 5 pathogens-12-00736-f005:**
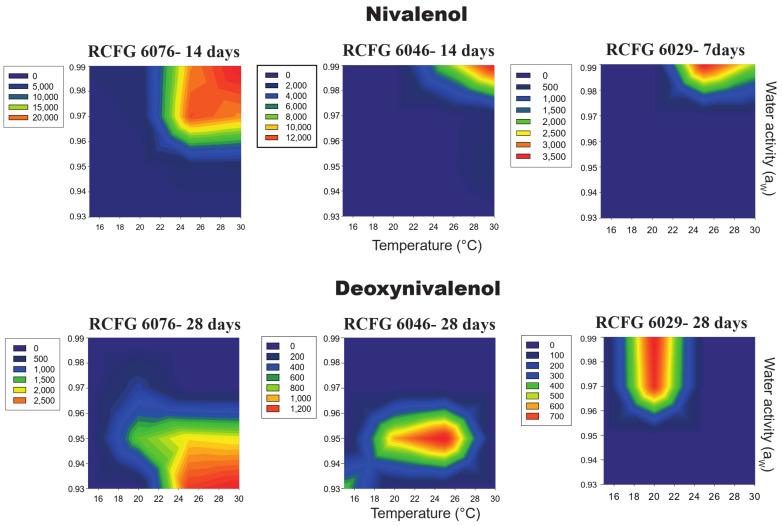
Two-dimensional contour maps of nivalenol and deoxynivalenol produced by three *Fusarium cerealis* strains (RCFG 6029, RCFG 6076, and RCFG 6046) according to water activity (a_W_) (0.93–0.99) and different temperatures (15 °C, 20 °C, 25 °C, and 30 °C) in a wheat-based medium at the incubation time when the maximum toxin levels were detected for each strain.

**Figure 6 pathogens-12-00736-f006:**
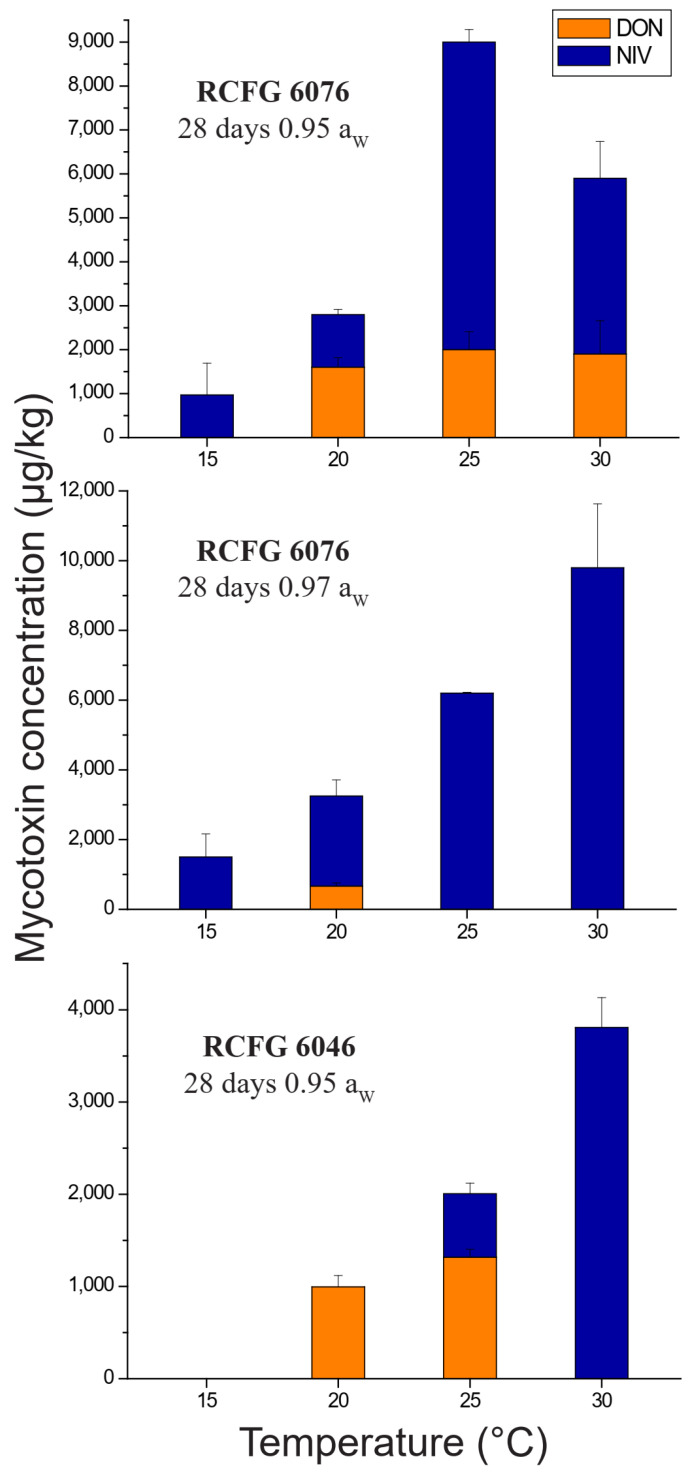
Nivalenol (NIV) and deoxynivalenol (DON) levels accumulated (µg/kg) produced by *Fusarium cerealis* RCFG 6076 and RCFG 6046 on wheat-based medium at different incubation conditions for water activity (a_W_) and temperature. Limit of detection (LOD): 15 µg/kg, Limit of quantification (LOQ): 40 µg/kg.

**Table 1 pathogens-12-00736-t001:** Analysis of variance (ANOVA) on the effects of strain (S), temperature (T°), water activity (a_W_), and their interactions on lag phase and growth rate of *Fusarium cerealis* in a wheat-based medium.

Source of Variation	Lag Phase (h)			Growth Rate (mm/day)		
	MS	F	*p*	MS	F	*p*
S	771.72	39.32	<0.0001	3.85	143.67	<0.0001
T°	1135.92	57.88	<0.0001	11.92	444.85	<0.0001
a_W_	16,896.93	860.95	<0.0001	244.77	9136.71	<0.0001
T° × a_W_	150.71	7.68	<0.0001	0.90	33.77	<0.0001
T° × S	210.78	10.74	<0.0001	0.26	9.6	<0.0001
a_W_ × S	58.92	3	<0.01	0.53	19.97	<0.0001
T° × a_W_ × S	125.84	6.41	<0.0001	0.12	4.36	<0.0001

MS, mean squares; F, F-value; *p*, *p*-value according to LSD test.

**Table 2 pathogens-12-00736-t002:** Analysis of variance (ANOVA) on the effects of strain (S), temperature (T°), water activity (a_W_), incubation time, and their interactions on nivalenol and deoxynivalenol production by three *Fusarium cerealis* strains (RCFG 6029, RCFG 6076, and RCFG 6046) in wheat-based media.

Source of Variation	Nivalenol		Deoxynivalenol	
	F	*p*	F	*p*
S	232.68	<0.0001	40.46	<0.0001
T°	53.86	<0.0001	6.55	0.0003
a_W_	197.17	<0.0001	94.07	<0.0001
Incubation time	14.38	<0.0001	165.91	<0.0001
T° × S	12.93	<0.0001	9.68	<0.0001
a_W_ × S	14.04	<0.0001	10.02	<0.0001
S x incubation time	7.76	<0.0001	9.32	<0.0001
T° × a_W_	4.58	<0.0001	13.82	<0.0001
T° × incubation time	5.54	<0.0001	2.42	0.0678
a_W_ × incubation time	14.19	<0.0001	26.63	<0.0001
S × T° × a_W_	4.03	<0.0001	5.31	<0.0001
S × T° × incubation time	3.18	<0.0001	11.9	<0.0001
S × a_W_ × incubation time	4.46	<0.0001	5.07	<0.0001
T° × a_W_ × incubation time	2.99	<0.0001	10.75	<0.0001
S × T° × a_W_ × incubation time	2.90	<0.0001	5.59	<0.0001

F, F-value; *p*, *p*-value according to LSD test.

## Data Availability

Not applicable.

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
