# Peer review of "Effect of Temperature, Water Activity and Incubation Time on Trichothecene Production by Fusarium cerealis Isolated from Durum Wheat Grains"

_pathogens, 2023, doi:10.3390/pathogens12050736_

Round 1

Reviewer 1 Report

The research manuscript details efforts by the authors to determine how environmental conditions – specifically water activity, temperature, and incubation time – impacts Fusarium cerealis mycotoxin production. The authors found significant interactions between strain, water activity, temperature, and incubation time. Each pathogen strain responded differently to growing conditions; each strain differed in growth, mycotoxin production and toxin profile.

The article is a relatively straight forward study on how growing conditions will impact the growth and mycotoxin production of a cereal pathogen Fusarium cerealis. The methods are appropriate and easily reproducible. The impact of the study is somewhat limited due to the high degree of variability observed between fungal strains coupled with the small number of strains tested.

Specific Comments:

Line 60: Please include additional details describing perithecia formation and the infection route into the plant. Temperature and precipitation are key factors in the likelihood of infection and would fit nicely with your research on the impact of temperature and water activity on pathogen growth.

Line 175 and 183: You state that within some aw, temperature showed little or no effect on growth rates, but only discuss this on line 183. Consider moving line 175 closer to 183 to improve the flow of the results.

Line 187: Please clarify this sentence construction, it currently links sentences 186 and 187 and implies that strain 6046 growth at 15°C and 0.93 aw is equivalent to the growth rates at 25-30°C and 0.95aw.   

Line 212-215: Should include statement describing how samples at lower temperatures and water activities had toxin concentrations below the limit of detection.

Lines 257-262: This does not belong in the discussion section, move to the introduction, or delete.

Figure 3 & 4: The axis font is too small. Figures are of poor resolution and the color scheme should match the Nivalenol and DON color profile of figure 6 (or vice versa). Panels for the temperature conditions that had no detectable mycotoxins should be included, even if they are flat planes.

Line 296-297: Lines should be rewritten to improve statement clarity.

Line 299: Why is day 14 the highest, was the NIV concentrations statistically equivalent at Day 21 & 28? Did NIV production cease and stop accumulating after day 14?

Line 319: Why does DON production only occur after 28 days of incubation? This should be addressed within the discussion.

Line 344: Why does RCFG 6076 produce far more toxin than the other strains? Why are there such significant differences between strains in terms of toxin profile and response to growing conditions? While you may not have the answers to these questions, they should at least be addressed within the discussion section.

Good overall quality of the manuscript, requiring a few sentence clarifications.

Author Response

Dear Reviewer

We sincerely appreciate your work and the helpful comments on our manuscript. Here our point by point responses .

The impact of the study is somewhat limited due to the high degree of variability observed between fungal strains coupled with the small number of strains tested.

Response: We think that the number of strains used in the present study was enough in order to cover most of the intraspecific variability that we can find within a species, specially within the genera Fusarium. As stated by Rabaaoui and co-workers Fusarium species are characterized by a wide inter- and intraspecific genetic diversity that is reflected in their high mycotoxin profile variability. The strains used in our study were selected over others because of their different mycotoxin production profile on durum wheat grain cultures.

Specific comments:

Line 60: Please include additional details describing perithecia formation and the infection route into the plant. Temperature and precipitation are key factors in the likelihood of infection and would fit nicely with your research on the impact of temperature and water activity on pathogen growth.

Response: New information was added in the introduction section as you suggested: “In the field, the Fusarium inoculum remains on crop debris as ascospores within perithecia (sexual structures) or macroconidia, and there it can resist the environmental conditions. Under favorable weather conditions (high relative humidity and warm temperatures) during wheat anthesis, the inoculum is capable of disperse through the wind or rain and reach the anther starting the plant infection. First, the spores germinate, the hyphae grow on the ovary, palea and lemma, and after that, they start the mycotoxin production.”

Line 175 and 183: You state that within some aw, temperature showed little or no effect on growth rates, but only discuss this on line 183. Consider moving line 175 closer to 183 to improve the flow of the results.

Response: The sentence “Within some aW, temperature showed little or no effect on growth rates.” Was moved as to this paragraph: “The highest growth rate was observed for RCFG 6076 and RCFG 6046 strains at 25 °C, 0.99 aW (8.27 and 8.14 mm/day, respectively), and the values were statistically significantly different compared to the rest of the growth rates obtained. Within some aW, temperature showed little or no effect on growth rates. At 25 - 30 °C there were not statistically significant differences between the growth rate at 0.93, 0.95 and 0.97 aW for strain RCFG 6076, at 0.95 and 0.97 for strain RCFG 6029 and at 0.95 for strain RCFG 6046.” 

Line 187: Please clarify this sentence construction, it currently links sentences 186 and 187 and implies that strain 6046 growth at 15°C and 0.93 aw is equivalent to the growth rates at 25-30°C and 0.95aw.

Response: The sentence was modified and part of the information was moved to the previous paragraph since it fits better. “At 25 - 30 °C there were not statistically significant differences between the growth rate at 0.93, 0.95 and 0.97 aW for strain RCFG 6076, and at 0.95 and 0.97 for strain RCFG 6029 and at 0.95 for strain RCFG 6046.”

Line 212-215: Should include statement describing how samples at lower temperatures and water activities had toxin concentrations below the limit of detection.

Response: A sentence was added as you suggested: “Before this period the toxin was not detected in samples from low aW and temperatures (LOD: 15 µg/kg, LOQ: 40 µg/kg)”

Lines 257-262: This does not belong in the discussion section, move to the introduction, or delete.

Response: The paragraph was removed from Discussion section: “Fusarium cerealis is a secondary pathogen involved in Fusarium Head Blight (FHB) of small grain cereals worldwide. However, recently, it has been reported as one of the main causal agents of the disease in barley, oats and wheat [17,18,20,22,23].” Was deleted while “Also, it is frequently isolated from maize where it can cause ear rot and stalk rot [42,43] and it was associated to root rot of soybean and ginseng [44,45].” Was moved to the Introduction section.

Figure 3 & 4: The axis font is too small. Figures are of poor resolution and the color scheme should match the Nivalenol and DON color profile of figure 6 (or vice versa). Panels for the temperature conditions that had no detectable mycotoxins should be included, even if they are flat planes.

Response: The figures 3 and 4 were modified according to your suggestions and also improved to increase resolution. Also, we added the LOD and LOQ in all corresponding figures.

Line 296-297: Lines should be rewritten to improve statement clarity.

Response: “Regarding mycotoxin analysis, the production patterns observed for the three F. cerealis strains were totally different, being strain RCFG 6076 the one that produced the highest mycotoxin levels. Nevertheless, it was possible to detect at least NIV in all conditions tested which coincide with those for growth.” Was re-written: “Regarding mycotoxin analysis, the production patterns observed for the three F. cerealis strains were totally different. Nivalenol was detected in all conditions in which F. cerealis was able to grow.”

Line 299: Why is day 14 the highest, was the NIV concentrations statistically equivalent at Day 21 & 28? Did NIV production cease and stop accumulating after day 14?

Response: “NIV production ranged from 41 to 23,918 µg/kg being the highest level detected at 0.99, 25 – 30 °C after 14 days of incubation. These temperatures and aW were the same for the maximum growth observed for F. cerealis.” Was re-written: “The highest levels were detected at 0.99, 25 – 30 °C between 14 and 21 days of incubation; after this period the levels decreased. These temperatures and aW were the same in which the maximum growth was observed for F. cerealis.”

Line 319: Why does DON production only occur after 28 days of incubation? This should be addressed within the discussion.

Response: we added a conclusion regarding this result: “the toxin was produced after 28 days of incubation by all strains. Why DON is produced after this period of time is a question that we still have to answer. This is to our knowledge the first ecophysiological study carried out about F. cerealis so we do not have other studies to compare. In order to answer this question, we are carrying out more studies, specially gene expression studies.”

Line 344: Why does RCFG 6076 produce far more toxin than the other strains? Why are there such significant differences between strains in terms of toxin profile and response to growing conditions? While you may not have the answers to these questions, they should at least be addressed within the discussion section.

Response: We addressed this result in the discussion section as you suggested: “In addition, the present study has shown that it is necessary to include more than one strain in the ecophysiological studies to obtain more reliable results due to the intraspecific variability that exists in a species, mainly within Fusarium [53]. For instance, in our study we observed that, regarding growth conditions, the three strains shared the same patterns. However, when we analyzed the mycotoxin production they had different production profiles. Strain RCFG 6076 produced the highest mycotoxin levels in all conditions. This is in accordance with our previous work where we observed that this strain produced the maximum NIV level on durum wheat grains cultures [27]. Also, we observed that strain RCFG 6029 produced DON in completely different conditions compared to the other strains. We could not have observed these results using only one strain.”

Good overall quality of the manuscript, requiring a few sentence clarifications.

Response: we tried to improve the clarity of the manuscript, specially the sentences you suggested.

Reviewer 2 Report

The study investigates the effect of environmental factors on the growth and mycotoxin production of Fusarium cerealis strains, which are causal agents of Fusarium Head Blight in wheat and produce both DON and NIV. The findings suggest that F. cerealis strains can grow in a wide range of conditions but mycotoxin production is strain and environment dependent. The study highlights the importance of considering the simultaneous presence of both toxins in grains, which could pose a more significant risk for contamination. However, there are suggestions to improve the clarity and flow of the information

  1. The objective of the study should be stated more explicitly in the abstract. Instead of "This study aimed to determine the effect...", it could be rephrased to "The objective of this study was to investigate the impact of environmental factors on the growth and mycotoxin production of F. cerealis strains."
  2. The findings of the study should be presented in a more organized manner. For instance, "F. cerealis strains were able to grow in a wide range of water activity and temperatures, but their mycotoxin production was influenced by strain and environmental factors. NIV was produced at high levels while optimal conditions for DON production were observed at low water activity. Interestingly, some strains were able to produce both toxins simultaneously, which could pose a more significant risk for grain contamination."
  3. Including Fusarium graminearum in the study would have allowed for a comparison of the effects of environmental factors on the growth and mycotoxin production of both species, providing a more comprehensive understanding of the factors that influence mycotoxin contamination in wheat. Additionally, as F. graminearum is another important causal agent of Fusarium Head Blight in wheat and also produces DON, it would have allowed for a comparison of mycotoxin production between the two species.
  4. Please italicize all the scientific names.
  5. Line 197-205 improve this paragraph for better clarity.
  6. Fig 1: Improve the figure quality. Correct the legends and denote C, D. Six templates in figure one but not clearly mentioned in the legends.
  7. Figure 3 quality is low due to which some comparative results are not observable. One suggestion is to use contrasting colour combinations for a better representation of the graphs.
  8. Line 295 onward paragraph requires re-writing.
  9. Conclusion should be re-written and avoid using citations in the conclusion. There should be a clear and concise conclusion that summarizes the main takeaways from the research.

Moderate English editing is required. 

Author Response

Dear Reviewer

We sincerely appreciate your work and the helpful comments on our manuscript. Here our point by point responses .

Comment 1: The objective of the study should be stated more explicitly in the abstract. Instead of "This study aimed to determine the effect...", it could be rephrased to "The objective of this study was to investigate the impact of environmental factors on the growth and mycotoxin production of F. cerealis strains."

Response: Thank you for the comment, we changed the objective of the study in the abstract as you suggested.

Comment 2: The findings of the study should be presented in a more organized manner. For instance, "F. cerealis strains were able to grow in a wide range of water activity and temperatures, but their mycotoxin production was influenced by strain and environmental factors. NIV was produced at high levels while optimal conditions for DON production were observed at low water activity. Interestingly, some strains were able to produce both toxins simultaneously, which could pose a more significant risk for grain contamination."

Response: thank you for the comment, we changed the results and conclusion of the abstract as you mentioned: “All strains were able to grow in a wide range of water activity (aW) and temperatures, but their mycotoxin production was influenced by strain and environmental factors. NIV was produced at high aW and temperatures while optimal conditions for DON production were observed at low aW. Interestingly, some strains were able to produce both toxins simultaneously, which could pose a more significant risk for grain contamination.”

Comment 3: Including Fusarium graminearum in the study would have allowed for a comparison of the effects of environmental factors on the growth and mycotoxin production of both species, providing a more comprehensive understanding of the factors that influence mycotoxin contamination in wheat. Additionally, as F. graminearum is another important causal agent of Fusarium Head Blight in wheat and also produces DON, it would have allowed for a comparison of mycotoxin production between the two species.

Response: we agree that it would be interesting for comparison but we did not include F. graminearum in the current study since Ramirez et al., (2006) evaluated F. graminearum strains isolated from wheat from Argentina. Also, in the discussion section we compared our results with this study and with other carried out by Belizan et al., (2019) who studied the ecophysiology of F. graminearum strains isolated also from Argentina. 

Comment 4: Please italicize all the scientific names.

Response: all scientific names were italicized as you suggested.

Comment 5. Line 197-205 improve this paragraph for better clarity.

Response: Paragraph “All the abiotic factors under study (aW, temperature and incubation time) and their interactions influenced NIV and DON production by the three F. cerealis strains on a wheat-based medium (Table 2). The analysis of variance of NIV production results showed that the most important factors that influenced the toxin production were strain, aW, and temperature. While for DON, the incubation time and aW were the main factors. The figures 3 and 4 show the NIV and DON production by F. cerealis strains at 15, 20, 25 and 30 °C, at four different aW, during 7, 14, 21 and 28 days of incubation. In general, F. cerealis was able to produce DON and NIV at all the conditions tested, however the amount and the type of toxin was variable. Studied strains showed different trichothecene production profiles.” Was re-written: “All the factors under study (aW, temperature, incubation time and strain) and their interactions influenced NIV and DON production by the three F. cerealis strains on a wheat-based medium (Table 2). The analysis of variance showed that the most important factor that influenced the NIV production was strain. While for DON production, the incubation time and aW were the most important. The figures 3 and 4 show the NIV and DON production by F. cerealis strains at 15, 20, 25 and 30 °C, at four different aW, during 7, 14, 21 and 28 days of incubation. In general, F. cerealis was able to produce DON and NIV at all the conditions tested, however the amount and the type of toxin was strain dependent since they showed different trichothecene production profiles.”

Comment 6. Fig 1: Improve the figure quality. Correct the legends and denote C, D. Six templates in figure one but not clearly mentioned in the legends.

Response: The quality of the figure was improved as you suggested. Also we added letters in the figure and their corresponding information in the legend. “Figure 1. Lag phases (A, B, C) and growth rates (D, E, F) of three Fusarium cerealis strains (RCFG 6029, RCFG 6076, and RCFG 6046) according to water activity (aW) (0.93- 0.99) and different temperatures (15, 20, 25, 30 °C) in a wheat-based medium. The error bars indicate the standard deviation for the triplicates. Different letters indicate statistical differences according LSD-test (p<0.05).”

Comment 7. Figure 3 quality is low due to which some comparative results are not observable. One suggestion is to use contrasting colour combinations for a better representation of the graphs.

Response: The quality of figure 3 was improved in order to be able to compare the results.

Comment 8. Line 295 onward paragraph requires re-writing.

Response: The paragraph “Regarding mycotoxin analysis, the production patterns observed for the three F. cerealis strains were totally different, being strain RCFG 6076 the one that produced the highest mycotoxin levels. Nevertheless, it was possible to detect at least NIV in all conditions tested which coincide with those for growth. NIV production ranged from 41 to 23,918 µg/kg being the highest level detected at 0.99, 25 – 30 °C after 14 days of incubation. These temperatures and aW were the same for the maximum growth observed for F. cerealis. While the minimum levels detected for NIV were at 0.93, 15 and 20 °C, after 28 days of incubation, which was the first time that the toxin was produced under this aW. Therefore, F. cerealis is capable of producing NIV at conditions suboptimal for growth but it takes a longer period of time. These same results were observed for other Fusarium species such as F. graminearum and F. culmorum, which produced higher amounts of trichothecene at the higher aW and temperatures analyzed [34,36]. For instance, Ramirez et al., [35] observed that F. graminearum produced the maximum amount of DON at 0.995 aW and 30 °C on irradiated wheat grains. For those NIV producing species such as F. poae, F. meridionale, F. culmorum and F. asiaticum it has been reported that optimum conditions for toxin production were between 20 - 30°C and 0.98 - 0.995 aW [33,36,47,48].” Was re-written: “Regarding mycotoxin analysis, the production patterns observed for the three F. cerealis strains were totally different. 

Nivalenol was detected in all conditions in which F. cerealis was able to grow. The highest levels were detected at 0.99, 25 – 30 °C between 14 and 21 days of incubation; after this period the levels decreased. These temperatures and aW were the same in which the maximum growth was observed for F. cerealis. While the minimum levels detected for NIV were at 0.93, 15 and 20 °C, after 28 days of incubation. Therefore, F. cerealis is capable of producing NIV at conditions suboptimal for growth but it takes a longer period of time. These same results were observed for other Fusarium species such as F. graminearum and F. culmorum, which produced higher amounts of trichothecene at the highest aW and temperatures analyzed [38,40]. For instance, in Argentina, Ramirez et al., [39] observed that F. graminearum produced the maximum amount of DON at 0.995 aW and 30 °C on irradiated wheat grains. For those NIV producing species such as F. poae, F. meridionale, F. culmorum and F. asiaticum it has been reported that optimum conditions for toxin production were between 20 - 30°C and 0.98 - 0.995 aW [37,40,47,48].”

Comment 9. Conclusion should be re-written and avoid using citations in the conclusion. There should be a clear and concise conclusion that summarizes the main takeaways from the research.

Response: The conclusion was re-written and part of the information was added to the discussion section and a new comment was added as suggestion from the other reviewer. The new conclusion is: “F. cerealis strains were capable of producing NIV in a wide range of temperatures and aW. High amounts of this toxin were produced at 0.99 aW and 25 – 30 °C which are the same environmental conditions for optimal growth. These conditions are those needed for Fusarium head blight development. So, in years with conducive conditions for the disease, grains could be contaminated with high NIV levels if they are infected by F. cerealis. Although DON production was not detected under these conditions, the risk of grain contamination with this mycotoxin could be under storage conditions since the optimal production seems to be at low aW. Also, the toxin was produced after 28 days of incubation by all strains. Why DON is produced after this period of time is a question that we still have to answer. This is to our knowledge the first ecophysiological study carried out about F. cerealis so we do not have other studies to compare. In order to answer this question, we are carrying out more studies, specially gene expression studies. 

In addition, two strains were able to produce both toxin simultaneously. Therefore, grains could end up contaminated with both DON and NIV if F. cerealis is present, posing a more significant risk for human and animal consumption.”

English comment: Moderate English editing is required.

Response: we tried to improve the English from the manuscript specially in those sentences you suggested to re-write. 

Round 2

Reviewer 2 Report

All the comments were properly addressed.